# Post-transcriptional control drives Aurora kinase A expression in human cancers

**Roberta Cacioppo**[1¤a]*, **Deniz Rad**[1¤b], **Giulia Pagani**[2], **Paolo Gandellini**[2], **Catherine Lindon**[1]*

1 Department of Pharmacology, University of Cambridge, Cambridge, United Kingdom, 2 Department of Biosciences, University of Milan, Milan, Italy

¤a Current address: MRC Laboratory of Molecular Biology, Cambridge, United Kingdom
¤b Current address: Trinity College, University of Cambridge, Cambridge, United Kingdom
* rcacioppo@mrc-lmb.cam.ac.uk (RC); acl34@cam.ac.uk (CL)

**Data Availability Statement:** The data underlying the results presented in the study are available in Supporting Information.

## Abstract

Aurora kinase A (AURKA) is a major regulator of the cell cycle. A prominent association exists between high expression of AURKA and cancer, and impairment of AURKA levels can trigger its oncogenic activity. In order to explore the contribution of post-transcriptional regulation to AURKA expression in different cancers, we carried out a meta-analysis of -omics data of 18 cancer types from The Cancer Genome Atlas (TCGA). Our study confirmed a general trend for increased AURKA mRNA in cancer compared to normal tissues and revealed that AURKA expression is highly dependent on post-transcriptional control in several cancers. Correlation and clustering analyses of AURKA mRNA and protein expression, and expression of AURKA-targeting *hsa-let-7a* miRNA, unveiled that *hsa-let-7a* is likely involved to varying extents in controlling AURKA expression in cancers. We then measured differences in the short/long ratio (SLR) of the two alternative cleavage and polyadenylation (APA) isoforms of AURKA mRNA across cancers compared to the respective healthy counterparts. We suggest that the interplay between APA and *hsa-let-7a* targeting of AURKA mRNA may influence AURKA expression in some cancers. *hsa-let-7a* and APA may also independently contribute to altered AURKA levels. Therefore, we argue that AURKA mRNA and protein expression are often discordant in cancer as a result of dynamic post-transcriptional regulation.

## Introduction

AURKA is a key regulator of the cell cycle, controlling centrosome maturation and mitotic spindle assembly [1]. AURKA is overexpressed and represents a marker of poor prognosis in a broad range of human malignancies [2–4]. Because of the significant association between high AURKA expression and cancer progression, poor prognosis, and drug resistance, AURKA is a preferred target for anti-cancer strategies, especially the use of small molecule kinase inhibitors [4, 5] and targeted proteolytic tools [6, 7]. However, even the most promising AURKA inhibitors are still under clinical studies and only alisertib has concluded phase III clinical assessment

**Funding:** We were funded by the Department of Pharmacology to Roberta Cacioppo and UKRI | Biotechnology and Biological Sciences Research Council (BBSRC) (BB/R004137/1) to Catherine Lindon. The funders had no role in study design, data collection and analysis, decision to publish, or preparation of the manuscript.

**Competing interests:** The authors have declared that no competing interests exist.

[4]. Furthermore, the existence of kinase-independent AURKA activity suggests that deregulation of expression may be sufficient to promote some of its oncogenic functions. Accordingly, anti-cancer strategies aimed at reducing AURKA expression levels have been shown to be effective in suppressing the carcinogenicity of AURKA [8–13].

The literature mainly reports that AURKA overexpression in cancers is due to increased gene copy number, transcription, or protein stability [14]. Various single nucleotide polymorphisms (SNPs) of the *AURKA* gene have also been associated with cancer development and susceptibility [15]. Additionally, certain microRNAs (miRNAs) control AURKA expression in cancers where its overexpression is a driving factor or a marker of poor prognosis [16–20]. However, overexpression of AURKA protein in cancer does not always correlate with the above mechanisms, suggesting that other deregulated post-transcriptional events may occur [21–24].

One important step in the maturation of mRNAs is the cleavage and polyadenylation (C/P) process, which involves cutting the 3' end of precursor mRNAs (pre-mRNAs) and adding a poly(A) tail [25]. A polyadenylation signal (PAS), which is found 10–30 nucleotides upstream, as well as UGUA and U-rich motifs usually precede the cleavage site, whereas U- and GU-rich motifs usually follow; these elements altogether form the C/P site [26]. Most human pre-mRNAs have multiple C/P sites [27], which allows for different transcript isoforms to be expressed for the same gene by alternative cleavage and polyadenylation (APA).

The function of microRNAs in controlling genes involved in the cell cycle and the significance of this control in cancer are widely recognised [28, 29]. The *hsa-let-7* miRNA family consists of 11 closely related genes that map in chromosomal areas that are usually deleted in human tumours [30]. Because of their pathogenic downregulation in cancer, *hsa-let-7* miRNAs are classified as tumour suppressors [28, 31]. A link between *hsa-let-7a* expression and clinical characteristics has been shown in triple negative breast cancer [32, 33], and roles for *hsa-let-7a* in breast tumour development and metastasis have been hypothesised [34, 35]. Multiple studies also reported a role for *hsa-let-7a* in controlling AURKA expression [19, 24, 31, 36]. Indeed, we recently described a post-transcriptional pathway of AURKA regulation through APA of AURKA mRNA that underlies cell cycle-dependent translational efficiency [24]. We found that perturbation of AURKA APA allowed AURKA expression to evade regulation by the tumour suppressor miRNA *hsa-let-7a* and was sufficient for acquisition of cellular properties associated with oncogenic transformation [24]. However, the extent of deregulated post-transcriptional events influencing AURKA expression across cancers is currently unknown.

Several studies have determined the expression profile of *AURKA* and its prognostic significance in a wide range of cancers using whole-genome datasets from TCGA [4, 37–41], although limited by two main caveats. First, these studies used incongruent datasets that included patients who had undergone prior treatments or who had been diagnosed with other types of cancers or metastases, as well as datasets that do not meet TCGA standards. In these studies cancer expression information was downloaded directly from user-friendly online resources, which perform downstream analysis of the entire group of datasets by TCGA, such as UALCAN [42] or GEPIA2 [43]. Second, the existence of multiple AURKA mRNA isoforms [44] was never considered. Given the key role of post-transcriptional control in cancer [45], analyses of isoform-specific expression may reveal mechanisms converging on mRNA that underlie dysregulated *AURKA* expression and that the studies carried out to date have overlooked.

In this article, analysis of protein and mRNA expression of *AURKA* in all cancers was performed using genomic datasets available from TCGA and explored in previous published studies [4, 37–40]. However, we improved the accuracy and reliability of the analysis by filtering datasets from patients with unusual medical histories or from those who have undergone

specific prior treatments, as well as non-standard datasets as determined by TCGA. Additionally, we sought to understand the role of post-transcriptional regulation in *AURKA* expression in cancer. Based on our previous finding that *hsa-let-7a* miRNA and APA of AURKA mRNA can control AURKA protein levels [24], we explored the extent to which such association may provide a general mechanism for AURKA overexpression in cancer. Based on our findings, we hypothesise that AURKA protein expression in cancer could result from dynamic post-transcriptional regulation by *hsa-let-7a* and APA, in combination or independently.

## Materials and methods

### Pan-cancer analysis of *AURKA* and *hsa-let-7a* expression

Public datasets of harmonized gene, miRNA and protein expression quantification of 18 primary tumour types from TCGA [41] (Table 1) were downloaded directly from the TCGA repository at https://portal.gdc.cancer.gov. Only solid tumours and tumours for which at least three datasets of matched normal tissues existed were considered for analysis. Datasets were excluded if they: (i) were derived from patients under systemic, radiation, neoadjuvant, hormone, or other treatments prior to the malignancy, or concomitant to the malignancy but claimed unsuitable by TCGA; (ii) were derived from patients previously or concomitantly diagnosed with other types of cancers or with metastases; (iii) were derived from patients with recurrent tumours; (iv) did not meet TCGA study protocols or did not have clear tumour classification. RStudio [46] was used to manage downloaded datasets and perform downstream analysis (S1 Script). Data points relative to AURKA mRNA and *hsa-let-7a* miRNA expression were extracted from transcriptome profiling and miRNA-seq datasets and plotted as fragments per kilobase of transcript per million mapped reads (FPKM) and as reads per million miRNA mapped (RPM), respectively, in $\log_2$ scale. Abundance of mature *hsa-let-7a* was calculated as the sum of reads from all miRNA IDs corresponding to the same unique MIMAT identifier (MIMAT0000062) from miRBase [47]. Data points of AURKA protein expression were extracted from proteome profiling datasets and plotted as $\log_2$ arbitrary units (a.u.). Bioinformatics pipelines used by TCGA to generate mRNA, miRNA, and protein expression quantification datasets can be found at docs.gdc.cancer.gov/Data/Introduction. Plots were created in GraphPad Prism.

### Correlational analysis

RStudio [46] was used to match variables to their originating sample according to the Sample ID for each TCGA cancer project (S1 Script). The variables were first matched in pairs; paired data points belonging to individual samples were then plotted in XY graphs to perform the Spearman's rank test, and the strength of correlation was expressed as value of the Spearman's rank correlation coefficient (r). Degrees of correlation were considered as follows: $r < |0.1|$ none; $r = [|0.1|\text{-}|0.3|]$ weak; $r = [|0.3|\text{-}|0.5|]$ moderate; $r = [|0.5|\text{-}|0.7|]$ strong; $r > |0.7|$ very strong. p values were corrected using the Bonferroni test in GraphPad Prism. Plots were created in GraphPad Prism.

### Clustering analysis

*k*-means clustering was used to partition two observations into clusters, in which each cancer belongs to the cluster with the nearest mean of the observations' values (cluster centroid) serving as a prototype of the cluster. The total sum of the squared distance between centroid and each member of the cluster (i.e., total SSE) was 0.2 and the SSE per cluster was $\leq 0.1$.

**Table 1. Comparison of the two pools of datasets for analysis of AURKA mRNA expression.**

| | Entire set | | | Filtered set | | | |
|---|---|---|---|---|---|---|---|
| | n° | Median FPKM | Median FPKM(T) / Median FPKM(NT) | n° | Median FPKM | Median FPKM(T) / Median FPKM(NT) | difference (%) |
| NT | 19 | 1.29 | 9.33 | 15 | 1.03 | 11.54 | 23.68 |
| BLCA | 412 | 12.04 | | 362 | 11.89 | | |
| NT | 113 | 0.93 | 7.88 | 110 | 0.93 | 7.91 | 0.41 |
| BRCA | 1118 | 7.33 | | 1090 | 7.36 | | |
| NT | 3 | 0.51 | 30.13 | 3 | 0.51 | 30.23 | 0.33 |
| CESC | 306 | 15.37 | | 303 | 15.42 | | |
| NT | 41 | 6.20 | 2.98 | 30 | 6.11 | 3.02 | 1.20 |
| COAD | 483 | 18.48 | | 408 | 18.43 | | |
| NT | 13 | 1.92 | 6.43 | 12 | 1.74 | 7.10 | 10.34 |
| ESCA | 185 | 12.36 | | 181 | 12.36 | | |
| NT | 44 | 2.78 | 4.08 | 39 | 2.68 | 4.19 | 2.72 |
| HNSC | 522 | 11.33 | | 483 | 11.22 | | |
| NT | 72 | 1.04 | 1.64 | 64 | 1.04 | 1.63 | -0.58 |
| KIRC | 542 | 1.71 | | 427 | 1.70 | | |
| NT | 32 | 1.00 | 1.99 | 28 | 0.98 | 2.06 | 3.58 |
| KIRP | 291 | 1.99 | | 261 | 2.02 | | |
| NT | 50 | 0.54 | 9.81 | 43 | 0.56 | 9.55 | -2.67 |
| LIHC | 374 | 5.30 | | 355 | 5.35 | | |
| NT | 59 | 1.21 | 5.69 | 46 | 1.16 | 5.98 | 5.07 |
| LUAD | 541 | 6.89 | | 462 | 6.94 | | |
| NT | 51 | 1.17 | 10.54 | 43 | 1.11 | 11.23 | 6.52 |
| LUSC | 502 | 12.33 | | 458 | 12.46 | | |
| NT | 4 | 1.53 | 2.41 | 4 | 1.52 | 2.46 | 2.30 |
| PAAD | 179 | 3.68 | | 157 | 3.74 | | |
| NT | 3 | 0.78 | 1.74 | 3 | 0.78 | 1.73 | -0.74 |
| PCPG | 184 | 1.36 | | 181 | 1.35 | | |
| NT | 52 | 0.63 | 1.76 | 44 | 0.60 | 1.87 | 5.95 |
| PRAD | 502 | 1.11 | | 475 | 1.12 | | |
| NT | 10 | 6.15 | 3.25 | 9 | 6.18 | 3.24 | -0.24 |
| READ | 167 | 19.97 | | 146 | 20.02 | | |
| NT | 36 | 2.82 | 4.80 | 36 | 2.82 | 4.77 | -0.52 |
| STAD | 412 | 13.53 | | 376 | 13.46 | | |
| NT | 59 | 0.86 | 0.89 | 57 | 0.86 | 0.89 | 0.16 |
| THCA | 513 | 0.77 | | 482 | 0.77 | | |
| NT | 35 | 0.67 | 11.91 | 34 | 0.69 | 11.45 | -3.87 |
| UCEC | 554 | 7.98 | | 521 | 7.90 | | |

For each pool (entire or filtered), the table displays the available number of RNA-seq datasets (n°), the median AURKA mRNA FPKM, and the ratio between tumour (T) to normal tissue (NT) of median FPKM, per cancer project and tissue type. The table also shows the % difference between the T / NT ratio when the filtered set is used and that when the entire set is used.

## Pan-cancer analysis of AURKA PAS usage

Mapped RNA-seq datasets in .bam format from the 18 selected TCGA solid primary tumours (Table 1) were obtained with permission of the NIH, and were analysed using the public software APAtrap [48] according to the developers' instructions. The analysis was only directed at *AURKA* and *CCND1* genes. GRCh38/hg38 was used as reference genome and the GENCODE

release 36 [49] was used for gene annotations. The analysis was run using APAtrap default parameters but with a minimum average coverage required for each 3'UTR of 10. Data relative to APA isoform usage of the genes of interest were plotted as short/long ratio (SLR) of individual samples, or as mean cancer to normal fold change of SLR values for individual normal-cancer sample pairs (i.e., SLR fold change). Plots were created in GraphPad Prism.

## Results

### Pan-cancer analysis of AURKA mRNA and protein expression

We first explored AURKA mRNA expression levels across cancers. Before performing a pan-cancer analysis of AURKA expression, we first explored whether the inclusion or exclusion of unusual samples affected fold change estimates between cancer and normal tissues. To do this, two pools of TCGA gene expression quantification datasets were created (Table 1). One contained the entire set of datasets from both cancer and normal samples as downloadable from TCGA (n = 8483). The second pool (n = 7748) excluded datasets from irregular cases, which were from both cancer and normal tissue samples. The difference between the fold change of median AURKA FPKM in cancer over normal samples when the entire pool is used (a) and when the filtered pool is used (b) was calculated as: difference (%) = [(a–b)/b] x 100. While this difference was < |5|% in 13 cancers, it was ~5–10% in four cancers (ESCA, LUSC, PRAD, LUAD), and ~24% in BLCA (Table 1). Differences were observed with both positive and negative trends, indicating that using nonconforming datasets can lead to both overestimate and underestimate of changes in AURKA mRNA expression between cancer and normal samples.

The pan-cancer analysis of AURKA mRNA expression revealed a general trend of increased mRNA levels in cancer tissues, although to varying extents (Fig 1A). Other studies reported different statistical significances due to the use of alternative statistical tests and a different number of control samples [4, 37–40]. THCA represents the only example in which AURKA mRNA is lower in cancer compared to normal tissue, although its expression is overall very low, and this difference is therefore non-significant (Table 1). Most cancer samples showed an extensive range of data distribution, indicating variable levels of AURKA mRNA expression among cancer patients. The median AURKA mRNA expression across normal tissues was tissue-dependent, and the overall AURKA mRNA levels in cancer also varied across tissues suggesting tissue-specific dysregulation.

Next, how AURKA mRNA expression correlated with protein expression ('protein vs mRNA') was examined in cancer samples (Figs 1B and S1). Proteomic datasets were also filtered for irregularities before analysis. For all the cancers, except PRAD and THCA, there existed a positive correlational relationship (r > 0.1) of varying degrees between AURKA mRNA and protein expression levels (Fig 1B). Degrees of correlation were considered as described in Materials and Methods. The wide range of r coefficient values suggests that mechanisms exist that modulate the coupling of AURKA mRNA and protein expression (Fig 1C). High r values would indicate that AURKA protein levels are dictated by mechanisms that determine the number of its mRNA molecules, such as gene copy number variation (CNV), the rate of transcription or of mRNA degradation. Contrarily, low r values might indicate that AURKA protein abundance likely results from mechanisms instead influencing the rate of translation or of protein degradation. Therefore, mechanisms governing AURKA translation or degradation play a greater role in cancers with lower AURKA mRNA-protein correlation (Fig 1C).

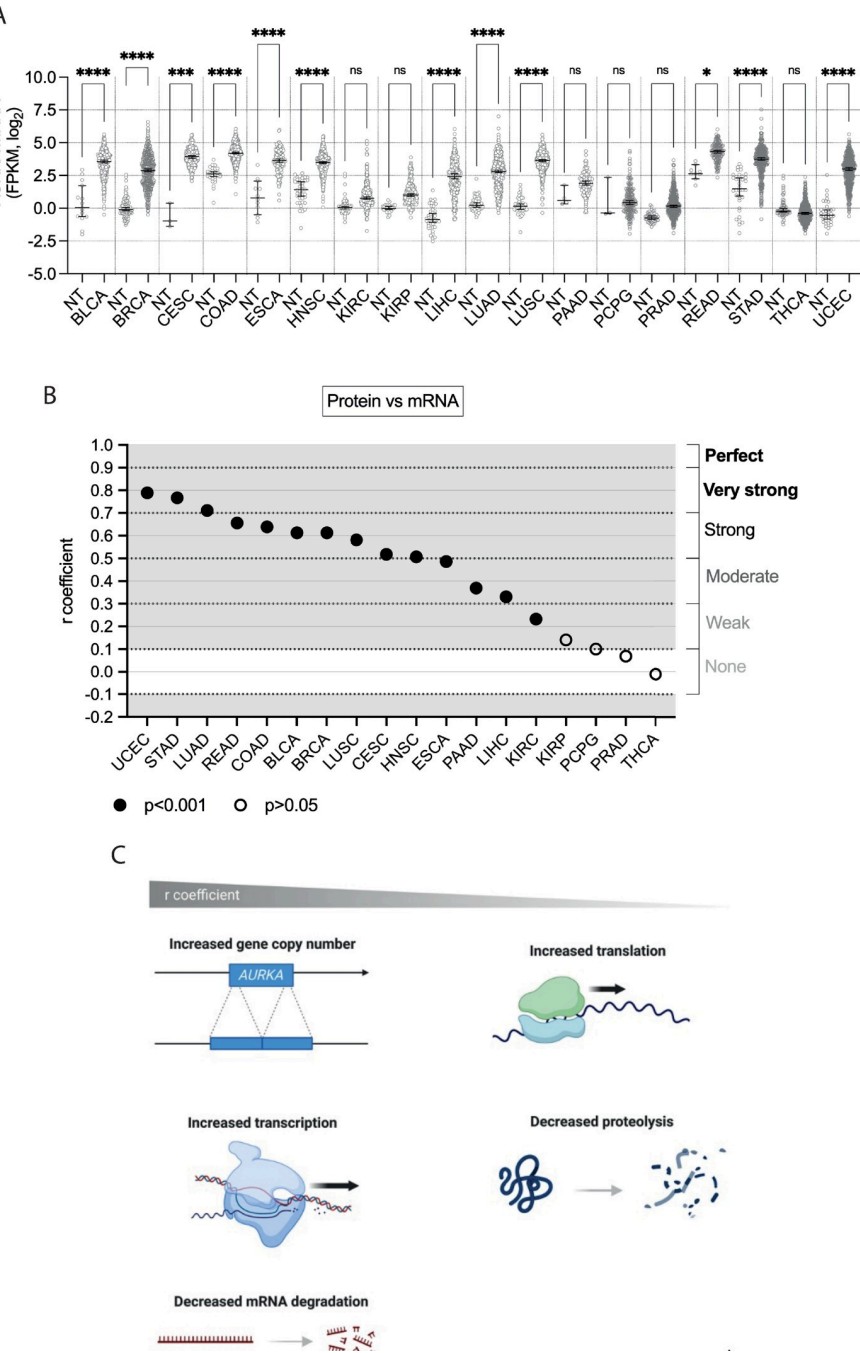

**Fig 1. Pan-cancer analysis of AURKA mRNA and protein expression. (A)** Median and 95% CI of AURKA mRNA
FPKM values in 18 TCGA cancers compared to their respective normal tissue (NT). Number of datasets per condition
shown in Table 1, filtered set. Kruskal-Wallis with Dunnett's multiple comparisons test. ns, not significant; *, p<0.02;
****, p<0.0001. **(B)** Distribution of r coefficients of the AURKA mRNA-protein correlation across cancers. p<0.001 for
all r coefficients except KIRP, PCPG, PRAD and THCA (p>0.05). **(C)** Graphic representation of molecular processes
likely underlying the AURKA mRNA-protein correlation in cancer samples. Figure created with BioRender.com.

## Pan-cancer analysis of *hsa-let-7a* miRNA expression

Since aberrant AURKA protein expression in cancer may derive from alterations in post-transcriptional events, this was further investigated in light of the reported role of *hsa-let-7a* miRNA in controlling AURKA expression [19, 24, 31, 36]. Initially, the selected 18 cancers were subjected to analysis of *hsa-let-7a* expression levels. Two pools of datasets were created, one containing the entire set of datasets from both cancer and normal samples as downloaded from TCGA (n = 8376), and a second one that excluded datasets pertaining to irregular cases (n = 7702) (Table 2). The percentage difference (%) between the fold change of median *hsa-let-7a* RPM in cancer over normal samples when the entire pool is used and when the filtered pool is used was calculated as per Table 1. While the difference was minimal for nine cancers (< |1|%), it was |1–5|% for six cancers (BLCA, BRCA, COAD, HNSC, LIHC, THCA), and ~| 6–9|% for three cancers (ESCA, KIRP, LUAD). Similarly to the analysis of AURKA mRNA expression, differences could be either positive or negative (Table 2).

The levels of *hsa-let-7a* expression in the selected cancers compared to corresponding normal tissues are shown in Fig 2A. Despite *hsa-let-7a*'s status as a known tumour suppressor miRNA [28, 31, 36], and therefore the expectation that its expression is down-regulated in cancer, five cancers (BLCA, CESC, ESCA, PAAD, PRAD) showed increased *hsa-let-7a* expression compared to normal tissues (Fig 2A and Table 2). Two cancers (STAD, UCEC) did not show any particular change in *hsa-let-7a* expression compared to their respective controls. On the other hand, 11 cancers showed a decrease in *hsa-let-7a* expression compared to normal tissues, which was statistically significant for all with the exception of KIRP. By examining *hsa-let-7a* expression in the normal samples across tissue types, it is evident that this varied considerably and was therefore tissue-dependent [30]. The general level of *hsa-let-7a* expression in cancer was also fluctuating across tissue types, implying tissue-specific mechanisms of dysregulation.

## Involvement of *hsa-let-7a* in AURKA expression across cancers

Subsequently, the correlation between AURKA mRNA and *hsa-let-7a* expression levels ('mRNA vs *hsa-let-7a*') in individual cancer samples was interrogated as a route to infer the role of *hsa-let-7a* in modulating AURKA expression (Figs 2B and S2A). Degrees of correlation were considered as described in Materials and Methods. A correlation was found to exist (r ≥ | 0.1|) with statistical significance for six cancers (BRCA, LUAD, LUSC, PRAD, THCA, UCEC). Nonetheless, in accordance with the role of *hsa-let-7a* in suppressing AURKA expression [19, 24, 31, 36], negative r coefficients were measured in all cases where r > |0.1|, except in PRAD, despite the general lack of statistical significance. A potentially existing correlation suggests that *hsa-let-7a* may act upon AURKA mRNA stability, whereas lack of correlation may not exclude that *hsa-let-7a* controls AURKA expression by tuning its translation, provided that a link exists between *hsa-let-7a* and AURKA protein expression.

The relationship between AURKA protein expression and *hsa-let-7a* expression ('protein vs *hsa-let-7a*') was then explored for the 18 cancers of interest (Figs 2C and S2B). An existing correlation (r ≥ |0.1|) with statistical significance was found for two cancers (BRCA, UCEC). In addition, negative r coefficients were measured for all cancers where r > |0.1|, with the exception of HNSC and PAAD. The correlations between *hsa-let-7a* and AURKA protein levels were overall weaker than those measured between *hsa-let-7a* and AURKA mRNA expression but might reflect a connection between *hsa-let-7a* and protein levels, underpinned by a role of *hsa-let-7a* in either AURKA mRNA abundance or translation rate, according to whether a link exists between *hsa-let-7a* and AURKA mRNA or not. A clustering analysis was therefore carried out between two chosen observations: r coefficients of the 'protein vs mRNA' correlation; r coefficients of the 'protein vs *hsa-let-7a*' correlation. Five clusters were identified (Fig 2D), the

**Table 2. Comparison of the two pools of datasets for analysis of *hsa-let-7a* expression.**

| | Entire set | | | Filtered set | | | |
|---|---|---|---|---|---|---|---|
| | n˚ | Median RPM | Median RPM(T) / Median RPM(NT) | n˚ | Median RPM | Median RPM(T) / Median RPM(NT) | difference (%) |
| NT | 57 | 5640 | 1.31 | 45 | 5780 | 1.25 | -4.02 |
| BLCA | 1254 | 7371 | | 1098 | 7250 | | |
| NT | 312 | 11684 | 0.82 | 306 | 11760 | 0.81 | -1.21 |
| BRCA | 3309 | 9528 | | 3192 | 9474 | | |
| NT | 9 | 7385 | 1.30 | 9 | 7385 | 1.29 | -0.54 |
| CESC | 927 | 9557 | | 897 | 9505 | | |
| NT | 24 | 18908 | 0.31 | 18 | 18841 | 0.31 | 2.36 |
| COAD | 1371 | 5803 | | 1158 | 5919 | | |
| NT | 39 | 7258 | 1.22 | 30 | 6788 | 1.30 | 6.73 |
| ESCA | 561 | 8864 | | 552 | 8848 | | |
| NT | 132 | 14321 | 0.65 | 114 | 14751 | 0.62 | -4.10 |
| HNSC | 1575 | 9241 | | 1464 | 9128 | | |
| NT | 213 | 9460 | 0.76 | 189 | 9460 | 0.76 | 0.06 |
| KIRC | 1635 | 7149 | | 1368 | 7153 | | |
| NT | 102 | 7486 | 0.83 | 87 | 8161 | 0.75 | -9.19 |
| KIRP | 876 | 6194 | | 786 | 6132 | | |
| NT | 150 | 12050 | 0.80 | 129 | 12519 | 0.78 | -2.73 |
| LIHC | 1125 | 9609 | | 1068 | 9710 | | |
| NT | 138 | 25917 | 0.35 | 99 | 27139 | 0.32 | -8.08 |
| LUAD | 1563 | 8974 | | 1344 | 8638 | | |
| NT | 135 | 13831 | 0.52 | 117 | 14195 | 0.52 | 0.64 |
| LUSC | 1434 | 7197 | | 1254 | 7434 | | |
| NT | 12 | 7594 | 1.31 | 12 | 7594 | 1.31 | 0.07 |
| PAAD | 537 | 9967 | | 456 | 9974 | | |
| NT | 9 | 18148 | 0.51 | 9 | 18148 | 0.51 | 0.07 |
| PCPG | 552 | 9179 | | 549 | 9185 | | |
| NT | 156 | 5440 | 1.75 | 132 | 5440 | 1.76 | 0.46 |
| PRAD | 1497 | 9507 | | 1416 | 9551 | | |
| NT | 9 | 13717 | 0.49 | 9 | 13717 | 0.50 | 0.79 |
| READ | 486 | 6748 | | 435 | 6801 | | |
| NT | 135 | 6823 | 1.07 | 135 | 6823 | 1.06 | -0.60 |
| STAD | 1338 | 7302 | | 1275 | 7258 | | |
| NT | 177 | 19193 | 0.76 | 171 | 19193 | 0.75 | -1.09 |
| THCA | 1542 | 14643 | | 1449 | 14483 | | |
| NT | 99 | 8123 | 1.05 | 99 | 8123 | 1.05 | 0.06 |
| UCEC | 1638 | 8549 | | 1635 | 8554 | | |

For each pool (entire or filtered), the table displays the available number of miRNA-seq datasets (n˚), the median *hsa-let-7a* RPM, and the ratio between tumour (T) to normal tissue (NT) of median RPM, per cancer project and tissue type. Note that the number of datasets in the entire pool reported in Table 2 refers to the total number of datasets before merging for the same unique MIMAT identifier. The table also shows the % difference between the T / NT ratio when the filtered set is used and that when the entire set is used.

members and features of which, as well as speculations on possible involvements of *hsa-let-7a* in AURKA expression, are summarized in Fig 2E. We point to a heterogeneous landscape of how *hsa-let-7a* might be implicated in the regulation of AURKA expression in cancer. Based on

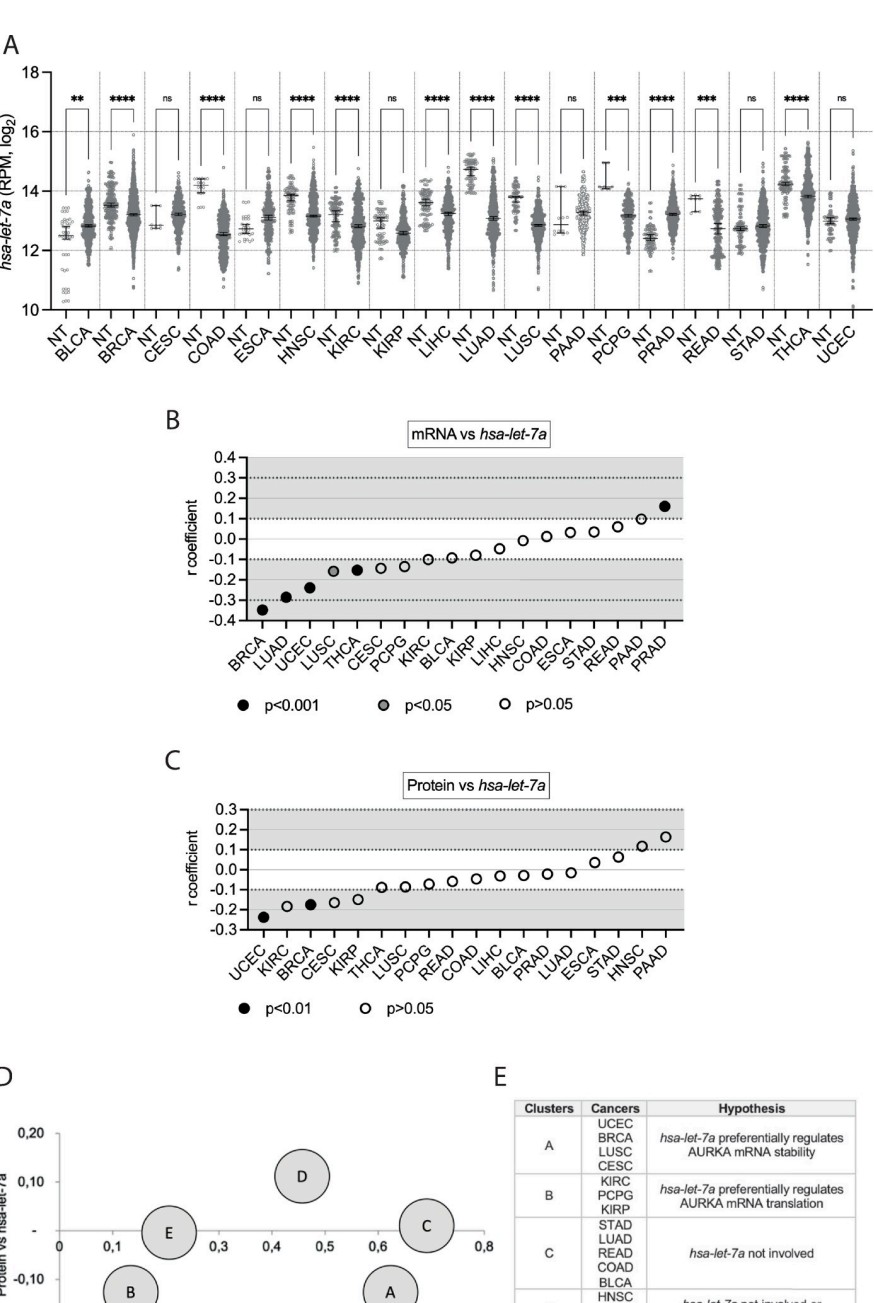

**Fig 2. Pan-cancer analysis of *hsa-let-7a* expression and involvement in AURKA expression. (A)** Median and 95% CI of *hsa-let-7a* RPM values in 18 TCGA cancers compared to their respective normal tissue (NT). Number of datasets per condition shown in **Table 2**, filtered set. Kruskal-Wallis with Dunnett's multiple comparisons test. ns, not significant; **, $p < 0.01$; ***, $p < 0.001$; ****, $p < 0.0001$. **(B)** Distribution of r coefficients of the AURKA mRNA-*hsa-let-7a* correlation across cancers. $p < 0.001$ for r coefficients of BRCA, LUAD, PRAD, THCA, UCEC; $p < 0.05$ for r coefficient of LUSC; $p > 0.05$ for all other r coefficients. **(C)** Distribution of r coefficients of the AURKA protein-*hsa-let-7a* correlation across cancers. $p < 0.001$ for r coefficients of BRCA and UCEC; $p > 0.05$ for all other r coefficients. **(D)** Clusters found according to common trends in the two types of correlation. The r coefficient of the indicated correlation is plotted on each axis. Data was also plotted on an interactive 3D graph visible in full interactive mode at math3d.org/bpYNRCnPA. **(E)** Table showing clusters composition and speculative involvement of *hsa-let-7a* in controlling AURKA expression.

our analysis, the involvement of *hsa-let-7a* could be hypothesised to range from being null to negatively controlling the rate of degradation or of translation of AURKA mRNA.

## Pan-cancer profiling of AURKA APA mRNA isoforms

One source of variability in the correlation between *hsa-let-7a* and AURKA expression in different cancers could arise from differences in AURKA mRNA processing that would determine sensitivity to *hsa-let-7a* targeting. For example, we previously showed that *hsa-let-7a* targets only the long APA isoform of AURKA mRNA [24] and a different short/long ratio (SLR) of AURKA APA isoforms is present between several types of normal and cancer cells [24, 50–53]. Here, pre-processed RNA-seq datasets from the selected TCGA cancers were subjected to analysis of PAS usage using APAtrap [48]. The APAtrap analysis only detected two of the AURKA transcripts annotated in ENSEMBL, namely ENST00000371356.6 (AURKA-203) and ENST00000395914.5 (AURKA-207). These transcripts share the same 3' UTR region, therefore the APAtrap results are identical, and we only report those for AURKA-203. The depth of coverage of reads mapped to the reference genome in is shown in Fig 3A for representative BRCA primary tumour samples, which show different AURKA SLRs. Most cancers showed an increase in mean AURKA SLR (AURKA-203) in cancer samples compared to respective matched normal samples (SLR fold change > 1) (Fig 3B). In most cases the cancer-dependent increase in mean SLR was more than 1.5-fold. In particular, BRCA showed an increase of ~2.5 fold change in mean SLR, and this recapitulates results that others have obtained using different methods [24, 50, 52]. As a positive control to our analysis, the APA-trap analysis was extended to the cyclin D1 (CCND1) mRNA, also known to have an increased SLR in cancer [54–57]. We found that the cancer/normal fold change of mean CCND1 SLR values was > 1 in most cases (Fig 3C), confirming the ability of APAtrap to report on APA events. In summary, our data indicate that APA represents a mechanism of AURKA post-transcriptional dysregulation. Given the higher translation rate of the short APA isoform [24], APA likely contributes to the increased expression of AURKA in cancers.

## Interplay of APA and *hsa-let-7a* in regulating AURKA expression

We then examined the extent to which APA is linked to AURKA expression and how it may be associated with *hsa-let-7a* in mediating AURKA expression across cancers. Degrees of correlation were considered as described in Materials and Methods. First, a positive correlation (r > 0.1) between AURKA SLR and protein expression levels ('protein vs SLR') was found in six cancers (Figs 4A and S3A), suggesting that APA modulates AURKA expression in these cancers (UCEC, BRCA, KIRP, LUAD, READ, KIRC). Furthermore, the 'protein vs SLR' correlation was analysed in light of the cancer-dependent changes in AURKA SLR (Fig 4B). A positive correlation was found (r = 0.2) when we estimated how the correlation of AURKA protein expression with its SLR varied according to SLR change in different cancers, although this has no statistical significance (p = 0.714). Our findings may potentially be consistent with the idea that APA positively contributes to shaping AURKA protein expression in certain cancers.

We then probed the correlation of AURKA SLR to AURKA mRNA expression ('mRNA vs SLR') across the six cancers (Figs 4C and S3B). Despite the degree of uncertainty in our measurements, these observations may be consistent with an effect of APA on AURKA mRNA stability in READ, BRCA, and UCEC, and lack of correlation in LUAD, KIRC, and KIRP instead possibly points to an effect on AURKA translation in these cancers. Finally, analysis of the correlation between AURKA SLR and *hsa-let-7a* expression levels ('hsa-let-7a vs SLR') suggests varying extents to which *hsa-let-7a* might be associated with AURKA APA. Different trends could be identified when AURKA 'protein vs SLR' correlation was compared to the 'hsa-let-7a

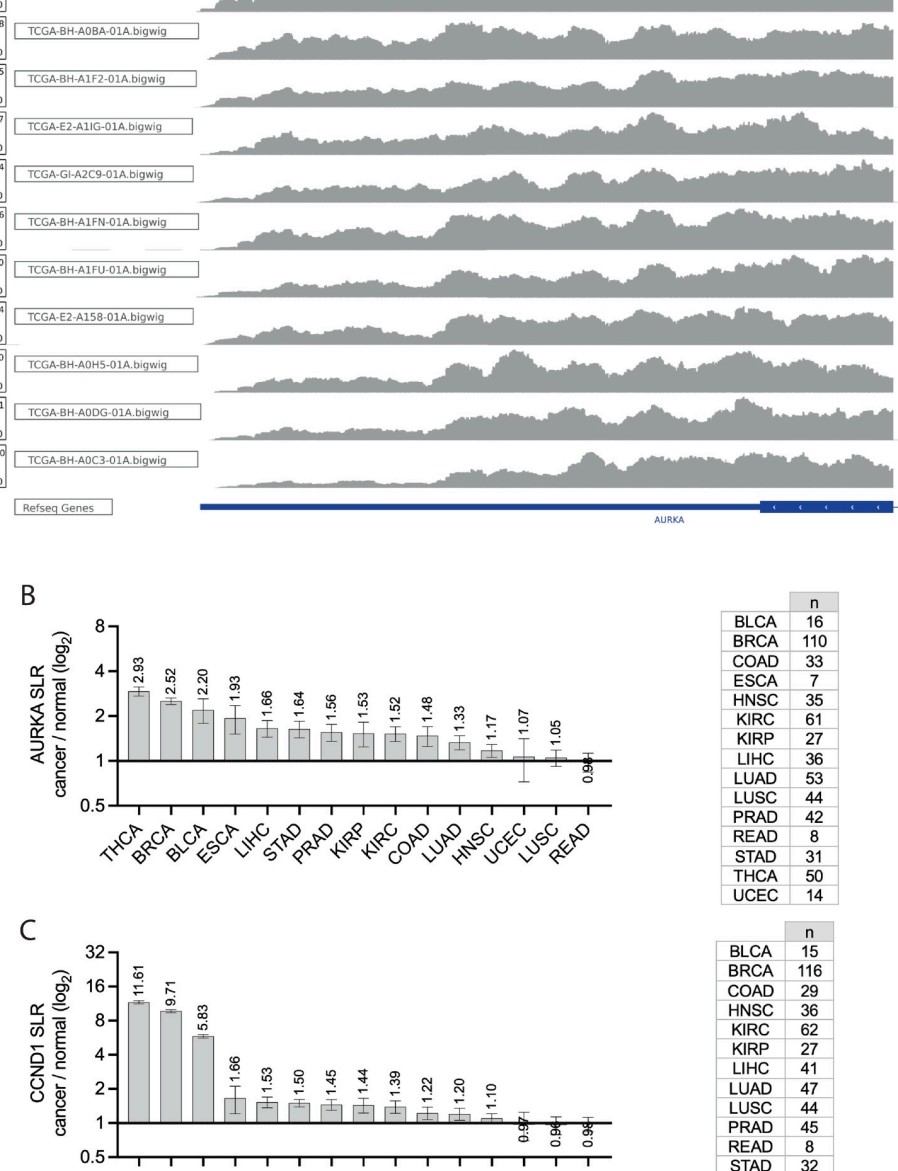

**Fig 3. Pan-cancer profiling of AURKA APA mRNA isoforms. (A)** Integrative Genomics Viewer (IGV) view of the depth of coverage of reads mapping to the 3' end of the AURKA-203 transcript (the last exon and 3' UTR are shown at the bottom, in blue). A subset of BRCA primary tumour samples is shown, selected to represent a range of SLRs, from low (top) to high (bottom). Reference genome GRCh38/hg38. Note, the y-axis scale (read depth) differs across panels. **(B)**, **(C)** Mean and standard error of the mean of SLR fold change values for cancer-normal sample pairs of AURKA transcript ENST00000371356.6 (B) and CCND1 transcript ENST00000227507.3 (C). Value of the mean indicated. SLR values were calculated using APAtrap and the analysis was performed once. n, number of cancer-normal sample pairs. Y axis in $\log_2$ scale.

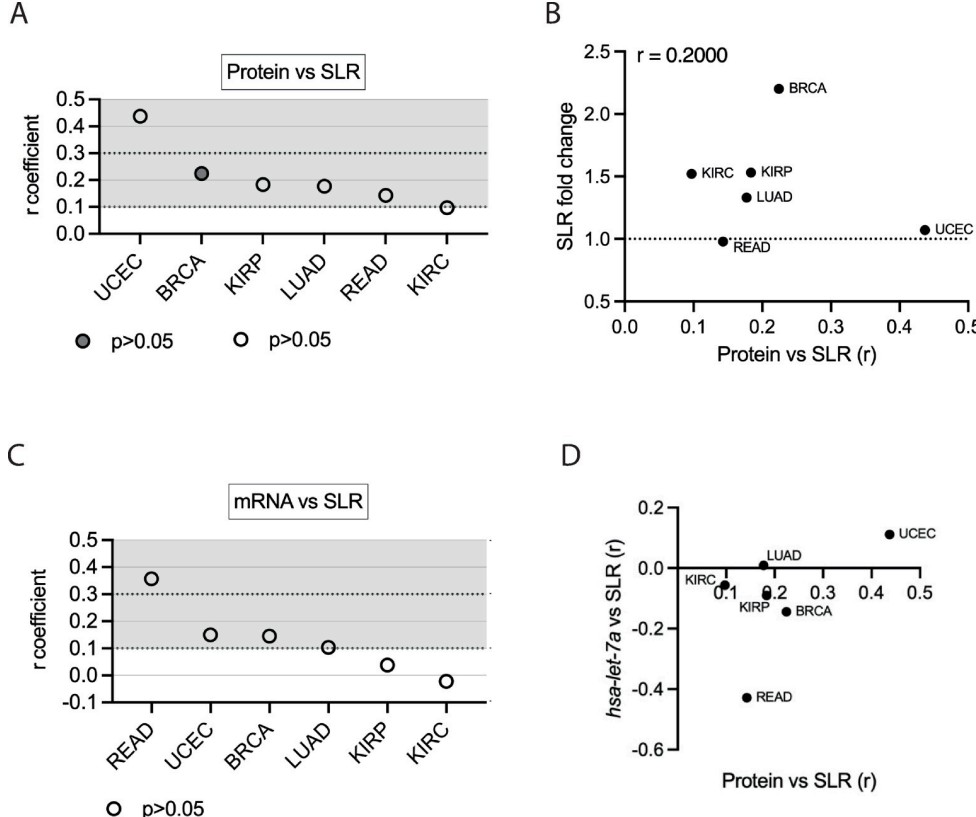

**Fig 4. Interplay of APA and *hsa-let-7a* in regulating AURKA expression. (A)** Distribution of r coefficients of the AURKA protein-SLR correlation across cancers. p<0.05 only for BRCA. **(B)** Scatter plot showing cancers according to the cancer/normal AURKA SLR fold change and to the value of the 'protein vs SLR' correlation coefficient r. **(C)** Distribution of r coefficients of the AURKA mRNA-SLR correlation across cancers. p>0.05 for all r coefficients. **(D)** Scatter plot showing cancers according to the r coefficient of the 'protein vs SLR' and of the '*hsa-let-7a* vs SLR' correlations.

vs SLR' correlation (Fig 4D). This raises the hypothesis that APA could influence AURKA expression via interplay with *hsa-let-7a* (UCEC, BRCA, KIRP, READ) or independently of *hsa-let-7a* (LUAD, KIRC).

## Heterogeneity of AURKA splicing isoforms across cancers

Seventeen high-scoring AURKA transcript variants have been annotated to date, nine of which are listed in the NCBI database [58] and eight in the Ensembl database [59] (Fig 5A). The 'matched annotation from NCBI and EMBL-EBI' (MANE) [60] label was recently given to two AURKA transcripts that were revealed identical (NM_198437.3 and ENST00000395915.8/AURKA-208). The 16 distinct isoforms are 5' untranslated region (UTR) splicing variants, resulting from alternative splice sites and exon skipping. The coding sequence and the 3'UTR instead follow canonical splicing, except for ENST00000395907.5 (AURKA-204), whose coding sequence includes the last intron and has a truncated 3'UTR; this transcript variant is however characterized by a poorer annotation score.

The public web server GEPIA2 [43] was consulted to retrieve information on the distribution of AURKA transcripts across TCGA cancers. Fig 5B shows the expression levels of each of the eight Ensembl transcripts in the selected TCGA cancers. It is likely that each isoform has an individual profile of expression across cancers, however a measure of the relative expression

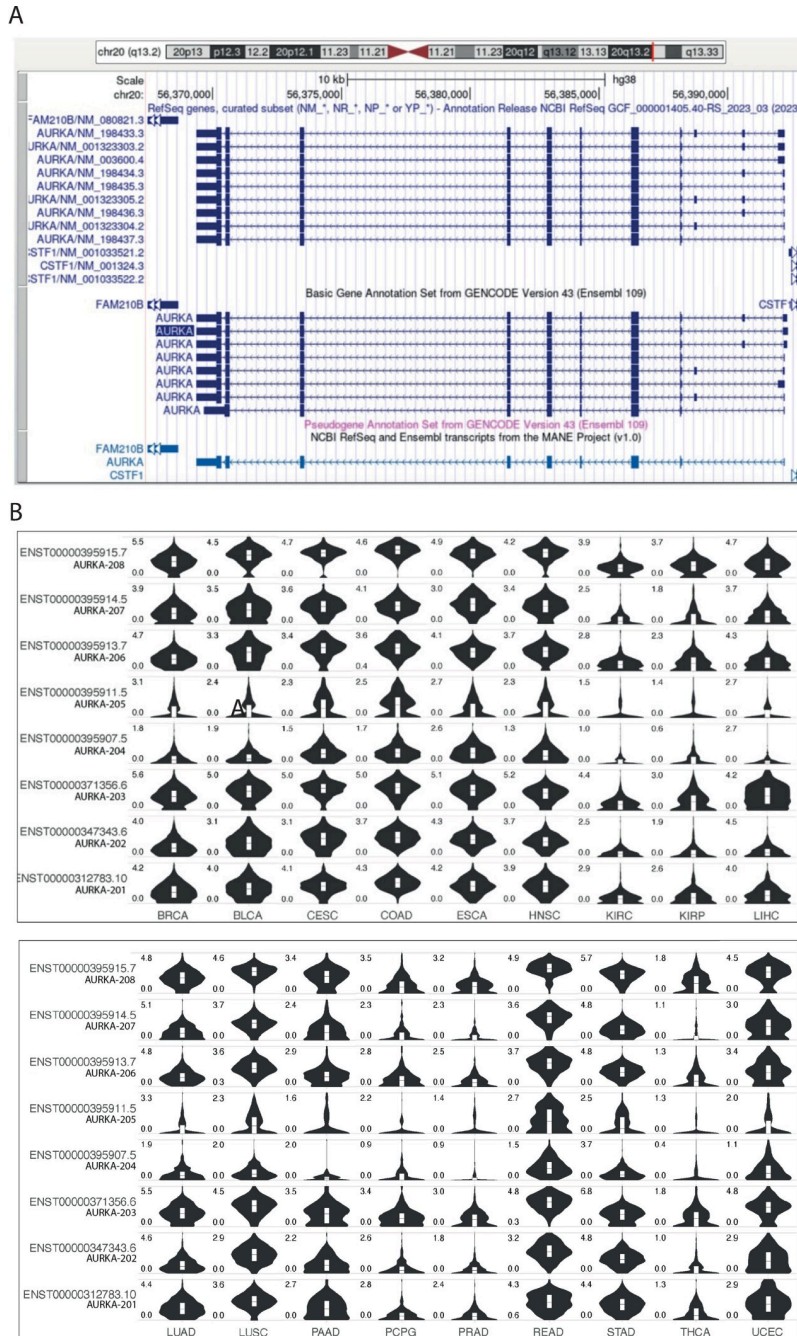

**Fig 5. Heterogeneity of AURKA splicing isoforms across cancers. (A)** UCSC Genome Browser view of AURKA transcript variants annotated in the NCBI (top) and Ensembl (bottom) databases. AURKA gene resides on the negative strand. The MANE isoform identical in both databases is shown in light blue at the bottom. **(B)** Violin plots showing the expression level [log2(TPM+1)] of the individual Ensembl AURKA transcripts in the TCGA cancers on the y axis. Cancer types in columns, AURKA transcripts in rows. TPM, transcript per million. Figure downloaded from the GEPIA2 platform. **(A)**, **(B)** ENST00000395907.5 (AURKA-204) transcript is displayed although it has lower annotation quality.

levels between the different isoforms would be required. In particular, some transcripts are abundantly expressed in most cancers, whereas others are expressed only in specific cancers.

Reciprocally, some cancers express most isoforms quite equally, while other cancers express only a specific group of transcripts. Since these are alternatively spliced isoforms in the 5'UTR with different combinations of exons [44] it can be concluded that splicing and unknown features of the 5'UTR play a role in determining the context-dependent expression profile of AURKA mRNA.

## Discussion

This article describes an exploratory analysis around the expression of *AURKA* on a pan-cancer scale. Following the analysis of publicly available data from TCGA, it was clear that *AURKA* overexpression at the level of the mRNA is prevalent in many cancers, a finding previously reported [4, 37–40]. However, our analyses of TCGA datasets avoided TCGA data-processing servers (e.g., UALCAN or GEPIA2). In addition, we carefully selected datasets to exclude those with annotations containing 'disqualifying' information about patients or samples. TCGA itself suggests reviewing datasets according to their accompanying annotations prior to running an analysis, although this practice is not commonly observed. Importantly, since a great number of flagged datasets derive from patients having received radiation, hormone, or other treatments, removing nonconforming datasets counteracts the bias in gene expression introduced by treatment-derived selective pressure, which constitutes a basis of tumour heterogeneity [61].

We investigated to what extent post-transcriptional regulation is responsible for defining and dysregulating *AURKA* expression in human cancers. To this aim, the extent to which the abundance of AURKA protein follows changes in the abundance of AURKA mRNA was measured across TCGA cancer samples. Our results suggest that mechanisms affecting AURKA translation or protein degradation are involved in tuning AURKA protein expression in various cancers, with a role of increasing importance as the degree of mRNA-protein correlation decreases. This has already been proposed as a general mechanism of gene expression regulation [62–64], and is concordant with results reported by others specifically regarding AURKA expression [21, 65–67].

Investigations of the possible involvement of *hsa-let-7a* miRNA in AURKA expression in cancer were also performed. First, analysis of expression levels of *hsa-let-7a* indicated that *hsa-let-7a* is downregulated in most cancers, in accordance with its tumour suppressor role [28, 31, 36], however with some exceptions. Such pan-cancer analysis of *hsa-let-7a* expression has no precedent in the literature to our knowledge. Our data support the idea that *hsa-let-7a* tunes the expression of AURKA in some cancers. Weak relationships were generally found across all the cancers, apparently inconsistent with a hypothesis that AURKA mRNA is a major target of *hsa-let-7a* [19, 31] unlike other oncogenes like RAS and MYC [68]. We hypothesise that mechanisms influencing the ability of *hsa-let-7a* to target AURKA mRNA could limit the coupling between *hsa-let-7a* abundance and AURKA expression measurable in our pan-cancer analysis. For example, *hsa-let-7a*'s targeting ability can indeed be subject to contextual regulation, such as cell cycle phase-dependent, since AURKA mRNA is specifically repressed by *hsa-let-7a* in $G_1$ and S phases but not in $G_2$ [24].

Other groups have used alternative software tools to measure genome-wide APA changes in TCGA samples, but in these studies AURKA SLR changes were not reported in the published lists of genes with cancer-dependent changes in SLR [69–74]. In contrast, other studies using alternative sample sources and methods of analysis found AURKA SLR increased in cancer samples, specifically in breast and lung cancers, where such increase even correlated with poor prognosis and survival [24, 50, 52]. Here, using the APAtrap software [48] to interrogate our curated TCGA samples, AURKA SLR was discovered to have increased in almost all

cancers compared to their normal tissues. Results presented here using high quality RNA-seq data are therefore consistent with what has previously been observed in breast and lung cancers using microarray data [24, 50, 52], and expand the observation to other cancers.

Overall, the diverse results on AURKA cancer-specific SLR across studies are not yet conducive to a unifying hypothesis for the APA-mediated regulation of AURKA. Interestingly, transcript AURKA-203 contains exon III that is thought to be implicated in breast cancer [75, 76], which is also the type of cancer showing the second highest change in AURKA SLR as recorded by the APAtrap analysis. Furthermore, while APAtrap exposes the abundance of one 3'UTR isoform compared to another of the same gene, abundance levels of each isoform can be otherwise influenced by mechanisms linked to alternative splicing. It is highly possible that there exists a combination of AURKA APA and alternative splicing isoforms in different contexts. Although there is no evidence to date that different AURKA transcripts might influence AURKA activity, instances of isoform-dependent protein localization and function are increasingly reported [77]. In a previous study, we have detected higher nuclear localization of a reporter protein under the regulation of AURKA short 3'UTR [24]. Therefore, there is a possibility that AURKA mRNA isoforms are targeted to different subcellular localizations to support localized translation–or that AURKA protein is co-translationally targeted to different compartments–and AURKA may be preferentially localized in the nucleus when coded by the short 3'UTR mRNA.

In conclusion, the data presented here cumulatively suggest that post-transcriptional mechanisms can constitute the primary form of regulation of the expression of AURKA in some cancers. As a consequence, impairments in mechanisms affecting mRNA dynamics offer a substantial basis for alterations in AURKA protein expression in cancer. This is a previously underrated notion, as the basis for AURKA oncogenic activation by means of overexpression has generally been attributed to enhanced gene copy number, transcription, or protein stability. We therefore add a new facet to the complex oncogenic expression program of the *AURKA* cancer gene, highlighting molecular mechanisms that could represent actionable targets of both DNA- and RNA-based therapeutics.

## Supporting information

**S1 Fig. Scatter plots displaying the correlation between AURKA mRNA and protein expression in 18 TCGA cancers.** Data points represent individual tumour samples. Values of the Spearman's rank coefficient (r) shown for each cancer. $p < 0.001$ for all r coefficients except KIRP, PCPG, PRAD and THCA ($p > 0.05$). Note that the ranges of the x and y axes are different in each panel.
(TIF)

**S2 Fig.** Scatter plots displaying the correlation between expression of AURKA mRNA and hsa-let-7a (A) or AURKA protein and hsa-let-7a (B) in 18 TCGA cancers. Data points represent individual tumour samples. Values of the Spearman's rank coefficient (r) shown for each cancer. Note that the ranges of the x and y axes are different in each panel. (A) $p < 0.001$ for r coefficients of BRCA, LUAD, PRAD, THCA, UCEC; $p < 0.05$ for r coefficient of LUSC; $p > 0.05$ for all other r coefficients. (B) $p < 0.001$ for r coefficients of BRCA and UCEC; $p > 0.05$ for all other r coefficients.
(ZIP)

**S3 Fig.** Scatter plots displaying the correlation between AURKA SLR and expression of AURKA protein (A) or AURKA mRNA (B) in selected TCGA cancers. Data points represent individual tumour samples. Values of the Spearman's rank coefficient (r) shown for each

cancer. Note that the ranges of the x and y axes are different in each panel. (A) $p < 0.05$ only for BRCA. (B) $p > 0.05$ for all r coefficients.
(ZIP)

**S1 Script. Code used for pan-cancer analyses of *AURKA* and *hsa-let-7a* expression and correlational analyses.**
(PDF)

**S1 Table. List of transcriptome profiling, miRNA-seq, proteome profiling, and RNA-seq datasets downloaded from TCGA used in this study.** The lists are provided as TCGA Manifest files for the straightforward download of publicly available datasets directly from TCGA.
(XLSX)

**S1 Data. Processed datasets underlying results of AURKA mRNA and *hsa-let-7a* expression, and AURKA SLR values.**
(ZIP)

**S2 Data. Processed datasets underlying results of correlational analyses.**
(XLSX)

## Acknowledgments

We thank members of the Lindon lab and Dr. Hugo Tavares for their input and discussions. The results published here are in whole or part based upon data generated by TCGA managed by the NCI and NHGRI. Information about TCGA can be found at http://cancergenome.nih.gov.

## Author Contributions

**Conceptualization:** Roberta Cacioppo.

**Data curation:** Roberta Cacioppo.

**Formal analysis:** Roberta Cacioppo, Deniz Rad.

**Funding acquisition:** Catherine Lindon.

**Investigation:** Roberta Cacioppo.

**Methodology:** Roberta Cacioppo, Deniz Rad, Giulia Pagani, Paolo Gandellini.

**Project administration:** Roberta Cacioppo.

**Resources:** Catherine Lindon.

**Supervision:** Catherine Lindon.

**Validation:** Roberta Cacioppo.

**Writing – original draft:** Roberta Cacioppo.

**Writing – review & editing:** Roberta Cacioppo, Giulia Pagani, Paolo Gandellini, Catherine Lindon.

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
