## [Decision Letter · Decision Letter 0]

29 Jul 2024

PONE-D-24-26023Post-transcriptional control drives Aurora kinase A expression in human cancersPLOS ONE

Dear Dr. Cacioppo,

Thank you for submitting your manuscript to PLOS ONE. After careful consideration, we feel that it has merit but does not fully meet PLOS ONE’s publication criteria as it currently stands. Therefore, we invite you to submit a revised version of the manuscript that addresses the points raised during the review process.

We look forward to receiving your revised manuscript.

Kind regards,

Chandrabose Selvaraj, Ph.D.

Academic Editor

PLOS ONE

Journal Requirements:

Department of Pharmacology:Roberta Cacioppo; UKRI | Biotechnology and Biological Sciences Research Council (BBSRC):Catherine Lindon BB/R004137/1

We thank members of the Lindon lab and Dr. Hugo Tavares for their input and discussions. RC is supported by a David James Studentship from the Department of Pharmacology and research in CL lab funded by Biotechnology and Biological Sciences Research Council [BB/R004137/1]. The results published here are in whole or part based upon data generated by TCGA managed by the NCI and NHGRI. Information about TCGA can be found at http://cancergenome.nih.gov

Department of Pharmacology:Roberta Cacioppo; UKRI | Biotechnology and Biological Sciences Research Council (BBSRC):Catherine Lindon BB/R004137/1

5. Thank you for uploading your study's underlying data set. Unfortunately, the repository you have noted in your Data Availability statement does not qualify as an acceptable data repository according to PLOS's standards.

7. We notice that your supplementary figures are included in the manuscript file. Please remove them and upload them with the file type 'Supporting Information'. Please ensure that each Supporting Information file has a legend listed in the manuscript after the references list.

Reviewers' comments:

Reviewer's Responses to Questions

**Comments to the Author**

1. Is the manuscript technically sound, and do the data support the conclusions?

Reviewer #1: Yes

Reviewer #2: Yes

Reviewer #3: Yes

2. Has the statistical analysis been performed appropriately and rigorously? 

Reviewer #1: Yes

Reviewer #2: Yes

Reviewer #3: Yes

3. Have the authors made all data underlying the findings in their manuscript fully available?

Reviewer #1: Yes

Reviewer #2: Yes

Reviewer #3: Yes

4. Is the manuscript presented in an intelligible fashion and written in standard English?

Reviewer #1: Yes

Reviewer #2: Yes

Reviewer #3: Yes

5. Review Comments to the Author

**Reviewer #1: **This study investigated the regulatory roles of hsa-let-7a miRNA and alternative cleavage and polyadenylation in AURKA overexpression in pan-cancers. In general, the results are clearly stated and the conclusions are reasonable based upon the presented data. Major comments raised by other reviewers have been answered well by the authors. In my opinion, the manuscript can be accepted at the present form.

**Reviewer #2: **The authors have provided satisfactory responses to the previously requested minor and major revisions. The exploration of AURKA mRNA and protein discordance, potentially due to dynamic post-transcriptional regulation, presents an interesting perspective that could be valuable to a wide range of researchers. Therefore, I recommend this work be considered for publication.

**Reviewer #3: **Summary

The manuscript investigates the post-transcriptional regulation of Aurora kinase A (AURKA) in human cancers through a meta-analysis of -omics data from The Cancer Genome Atlas (TCGA). The study highlights the role of alternative polyadenylation (APA) and hsa-let-7a miRNA in modulating AURKA expression and suggests their combined or independent influence on AURKA levels in cancer. The study's findings underscore the complexity of post-transcriptional regulation in cancer and propose potential molecular mechanisms for AURKA's oncogenic activation. Overall, the manuscript presents significant findings on the post-transcriptional regulation of AURKA in cancer. Addressing the above comments could further enhance the clarity, impact, and applicability of the research.

Major Comments

1. Methodological Transparency and Data Accessibility: The manuscript would benefit from greater transparency in the data preprocessing and analysis methods. Detailed descriptions of the bioinformatics pipelines, algorithms, and parameters used should be included (if possible, in the supplementary files). Additionally, providing access to the source code on platforms like GitHub and ensuring all processed datasets are available in a public repository such as Zenodo would facilitate reproducibility and independent validation of the results.

2. Gene Expression Regulation Analysis: Besides exploring miRNA's role in regulating AURKA expression, the authors should consider investigating the methylation status of the transcription start sites (TSS) of AURKA genes. Methylation, particularly hyper- and hypo-methylation at TSS, can significantly affect transcription. The TCGA and cBioPortal databases provide methylation profiles for these samples, which could be analysed to understand how methylation impacts AURKA mRNA levels. This additional analysis could offer further insights into the regulatory mechanisms influencing AURKA expression, complementing the findings on miRNA regulation

Figures and Data Presentation

Figure 1: I recommend plotting the data across multiple grids or rows to enhance the clarity and interpretability of Figure panel A, showing AURKA mRNA expression across various cancer types. This approach would make the patterns more discernible. For instance, arranging the data in three rows could help distinguish the differences more clearly and reduce visual clutter. Furthermore, including sample sizes for each group (cancer and normal tissues) in the figure legend would enhance interpretability.

The correlation coefficients (r) are well-displayed, but additional context on their statistical significance would strengthen the conclusions. For example, the dots or circles representing data points could be coloured according to the strength of the -log of the p-values. This approach would visually indicate the significance of each correlation, making it easier to identify which relationships are statistically robust. Additionally, providing a colour legend to interpret the p-value ranges would enhance the figure's clarity and utility, allowing readers to assess the reliability of the observed correlations quickly.

Figure 2: Analysis of hsa-let-7a miRNA expression: The analysis of hsa-let-7a expression across cancers is informative, but the figure could be improved as suggested for Figure 1 panels A and B. Additionally, the visual representation of the clustering analysis could be enhanced by labelling clusters with their respective cancer types for more straightforward interpretation.

Figure 4: The correlation coefficients (r) are well-displayed, but additional context on their statistical significance would strengthen the conclusions. For example, the dots or circles representing data points could be coloured according to the strength of the -log of the p-values.

Supplementary Figures: The figures showing the scatter plots should include both the R-square and the p-value.

6. PLOS authors have the option to publish the peer review history of their article (what does this mean?). If published, this will include your full peer review and any attached files.

Reviewer #1: No

Reviewer #2: No

Reviewer #3: No

---

## [Author Response · Author response to Decision Letter 0]

3 Sep 2024

Response:

We would like to thank the reviewers for their attentive reading of our manuscript. We appreciate all the comments and suggestions. We have addressed all the concerns and have included point-by-point responses labeled as green text in this document. In addition, we have addressed all the concerns of the reviewers from Review Commons according to our Revision Plan.

Please note that reference #77 (Mitschka, S., Mayr, C. Context-specific regulation and function of mRNA alternative polyadenylation. Nat. Rev. Mol. Cell. Biol. 23, 779–796 (2022)) was added to the text and references list.

Reviewer #3: 

Summary

The manuscript investigates the post-transcriptional regulation of Aurora kinase A (AURKA) in human cancers through a meta-analysis of -omics data from The Cancer Genome Atlas (TCGA). The study highlights the role of alternative polyadenylation (APA) and hsa-let-7a miRNA in modulating AURKA expression and suggests their combined or independent influence on AURKA levels in cancer. The study's findings underscore the complexity of post-transcriptional regulation in cancer and propose potential molecular mechanisms for AURKA's oncogenic activation. Overall, the manuscript presents significant findings on the post-transcriptional regulation of AURKA in cancer. Addressing the above comments could further enhance the clarity, impact, and applicability of the research.

Major Comments

1. Methodological Transparency and Data Accessibility: The manuscript would benefit from greater transparency in the data preprocessing and analysis methods. Detailed descriptions of the bioinformatics pipelines, algorithms, and parameters used should be included (if possible, in the supplementary files). Additionally, providing access to the source code on platforms like GitHub and ensuring all processed datasets are available in a public repository such as Zenodo would facilitate reproducibility and independent validation of the results.

Response:

For greater transparency, we have included the following files as Supporting Information (given file sizes less than 10MB): 

• Code used for analysis of AURKA mRNA and hsa-let-7a expression and correlational analyses (S1 Script); 

• The full list of transcriptome profiling, miRNA-seq, proteome profiling, and RNA-seq datasets downloaded from TCGA used in this study (S1 Table). The lists are provided as TCGA Manifest files for the straightforward download of publicly available datasets directly from TCGA; 

• Processed datasets underlying results in this study (S1 Data, S2 Data);

The source code for analysis of PAS usage is publicly available since it was developed and published by Ye et al., 2018. Unless otherwise stated, we have used the APAtrap code according to the developers’ instructions and code parameters used to suit our analysis are described in Materials and Methods.

2. Gene Expression Regulation Analysis: Besides exploring miRNA's role in regulating AURKA expression, the authors should consider investigating the methylation status of the transcription start sites (TSS) of AURKA genes. Methylation, particularly hyper- and hypo-methylation at TSS, can significantly affect transcription. The TCGA and cBioPortal databases provide methylation profiles for these samples, which could be analysed to understand how methylation impacts AURKA mRNA levels. This additional analysis could offer further insights into the regulatory mechanisms influencing AURKA expression, complementing the findings on miRNA regulation.

Response:

The methylation status of the transcription start sites (TSS) of AURKA gene is clearly an interesting area of investigation given the role of genetic and epigenetic alterations known to disrupt AURKA transcription. Although it is highly likely that hyper- and hypo-methylation at AURKA TSS impact AURKA mRNA levels, this kind of analysis is beyond the scope of this study, which is aimed at exploring the sole contribution of post-transcriptional regulation to AURKA protein expression in different cancers. In addition, although we agree that investigation of the methylation status of AURKA promoter would potentially enrich the study, this would be a new project that we do not currently have resources for and we are not in a position to carry out such analyses within the scope of this study.

Figures and Data Presentation

Figure 1: I recommend plotting the data across multiple grids or rows to enhance the clarity and interpretability of Figure panel A, showing AURKA mRNA expression across various cancer types. This approach would make the patterns more discernible. For instance, arranging the data in three rows could help distinguish the differences more clearly and reduce visual clutter. Furthermore, including sample sizes for each group (cancer and normal tissues) in the figure legend would enhance interpretability.

The correlation coefficients (r) are well-displayed, but additional context on their statistical significance would strengthen the conclusions. For example, the dots or circles representing data points could be coloured according to the strength of the -log of the p-values. This approach would visually indicate the significance of each correlation, making it easier to identify which relationships are statistically robust. Additionally, providing a colour legend to interpret the p-value ranges would enhance the figure's clarity and utility, allowing readers to assess the reliability of the observed correlations quickly.

Figure 2: Analysis of hsa-let-7a miRNA expression: The analysis of hsa-let-7a expression across cancers is informative, but the figure could be improved as suggested for Figure 1 panels A and B. Additionally, the visual representation of the clustering analysis could be enhanced by labelling clusters with their respective cancer types for more straightforward interpretation.

Figure 4: The correlation coefficients (r) are well-displayed, but additional context on their statistical significance would strengthen the conclusions. For example, the dots or circles representing data points could be coloured according to the strength of the -log of the p-values.

Response:

We thank the reviewer for advice on how to increase the clarity of the figures. We have addressed the comments as follows: 

• Figures 1A, 2A: we have included multiple rows in the plot

• Figures 1B, 2B, 2C, 4A, 4C we have color-coded the data points according to the p-value of the r coefficients 

• The visual representation of the clustering analysis in Figure 2D was enhanced by providing a link for an interactive 3D graphical representation, as was suggested by another reviewer

Supplementary Figures: The figures showing the scatter plots should include both the R-square and the p-value.

Response:

We are only using the Spearman’s rank correlation coefficient (r) and not the R-square value because the former can detect non-linear associations, while linear regression cannot and may skew results or lead to false conclusions.

---

## [Editor Report · Decision Letter 1]

5 Sep 2024

Post-transcriptional control drives Aurora kinase A expression in human cancers

PONE-D-24-26023R1

Dear Dr. Cacioppo,

We’re pleased to inform you that your manuscript has been judged scientifically suitable for publication and will be formally accepted for publication once it meets all outstanding technical requirements.

Kind regards,

Chandrabose Selvaraj, Ph.D.

Academic Editor

PLOS ONE
---

## [Editor Report · Acceptance letter]

6 Sep 2024

PONE-D-24-26023R1 

PLOS ONE

Dear Dr. Cacioppo, 

I'm pleased to inform you that your manuscript has been deemed suitable for publication in PLOS ONE. Congratulations! Your manuscript is now being handed over to our production team.

Kind regards, 

on behalf of

Dr. Chandrabose Selvaraj 

Academic Editor

PLOS ONE